# Health-related quality of life and hypertension in people with HIV on long-term antiretroviral therapy in Uganda

**Charles Batte[1]\*, Andrew Weil Semulimi[1], John Mukisa[2], Mariam Nakabuye[1], Jasper Nidoi[1], David Mukunya[3], Rosalind Parkes Ratanshi[4,5], Barbara Castelnuovo[4], Mohammed Lamorde[4], David Meya[6], William Checkley[7], Robert Kalyesubula[8], Trishul Siddharthan[9], Joseph B. Babigumira[10]**

1 Department of Medicine, Lung Institute, School of Medicine, College of Health Sciences, Makerere University, Kampala, Uganda, 2 Department of Immunology and Molecular Biology, School of Biomedical Sciences, College of Health Sciences, Makerere University, Kampala, Uganda, 3 Department of Community and Public Health, Faculty of Health Sciences, Busitema University, Tororo, Uganda, 4 Infectious Diseases Institute, Makerere University, Kampala, Uganda, 5 Department of Psychiatry, University of Cambridge, Cambridge, United Kingdom, 6 Department of Medicine, School of Medicine, College of Health Sciences, Makerere University, Kampala, Uganda, 7 Center for Global Non-Communicable Diseases Research and Training, Department of Medicine, Johns Hopkins University, Baltimore, Maryland, United States of America, 8 Department of Physiology, School of Biomedical Sciences, College of Health Sciences, Makerere University, Kampala, Uganda, 9 Department of Medicine, Miller School of Medicine, University of Miami, Miami, Florida, United States of America, 10 Saw Swee Hock School of Public Health, National University of Singapore, Singapore, Singapore

\* dr.cbatte@gmail.com

**Data Availability Statement:** The datasets supporting the conclusions of this article are available in the Figshare repository at the following

## Abstract

### Introduction

The presence of hypertension could reduce the health-related quality of life (HRQoL) of people with HIV (PWH). Yet, literature describing the HRQoL of PWH who have hypertension in Uganda is scarce making the design of locally adapted interventions cumbersome. In our study, we compared HRQoL scores of people with HIV with and without hypertension on long term antiretroviral therapy (ART) in Uganda.

### Methods

We recruited 149 PWH with hypertension and 159 PWH without hypertension in the long-term ART cohort at an urban clinic in Kampala, Uganda. Data on socio-demographics were collected using an interviewer designed questionnaire while data on the World Health Organisation clinical stage viral load and CD4 count as well as ART duration were extracted from clinic electronic database and a generic EuroQol -5D- 5L (EQ-5D- 5L) and Medical Outcome Study (MOS-HIV) questionnaire used to collect HRQoL data. Data were summarized using descriptive statistics while inferential statistics were used to determine associations between key variables and HRQoL. Mann-Whitney U tests were used to compare HRQoL between groups of interest.

link: https://doi.org/10.6084/m9.figshare.20014799.v1.

**Funding:** This project was supported by NIH Research Training Grant # D43 TW009345 funded (CB) by the Fogarty International Center, the NIH Office of the Director Office of AIDS Research, the NIH Office of the Director Office of Research on Women's Health, The National Heart, Lung and Blood Institute, the National Institute of Mental Health, and the National Institute of General Medical Sciences. The content is solely the responsibility of the authors and does not necessarily represent the official views of the National Institute of Health. The ALT cohort (BC) is funded by Janssen, the pharmaceutical company of Johnson & Johnson, through a grant to the Academy for Health Innovation, Uganda at Infectious Diseases Institute. The funders had no role in study design, data collection and analysis, decision to publish, or preparation of the manuscript.

**Competing interests:** The authors have declared that no competing interests exist.

**Abbreviations:** ALT, ART Long Term; ART, Antiretroviral Therapy; EQ- 5D- 5L, European Quality of Life 5 Dimensions 5 levels; HRQoL, Health Related Quality of Life; MOS-HIV, Medical Outcomes Study HIV Health Survey; NCDs, Non-Communicable Diseases; WHO, World Health Organization.

## Results

One hundred ninety (61.7%) participants were female. PWH who had hypertension were older (Mean ± SD: 53.7 ± 8.3 vs 49.9 ± 8.6, p value <0.001) than those without hypertension. Participants with hypertension had lower overall median health utility scores (0.71 (0.33–0.80) vs 0.80 (0.44–0.80), p value = 0.029) and mean physical health score (48.44 ± 10.17 vs 51.44 ± 9.65, p value < 0.001) as opposed to those without hypertension. Hypertension (p value = 0.023), high income status, >70,000 UGX, (p value = 0.044), disclosure of the HIV status of the participants to their partner (p value = 0.026), and current history of smoking (p value = 0.029) were associated with low HRQoL scores.

## Conclusion

Among people with HIV, those with hypertension had lower HRQoL compared to those without. This calls for inclusion of quality-of-life assessment in the management of PWH who have been diagnosed with hypertension to identify those at risk and plan early interventions.

## Introduction

The scaling up of lifesaving antiretroviral therapy (ART) has led to a significant reduction in AIDS related mortality from 1.4 million in 2010 to 630,000 in 2022 [1] with sub-Saharan Africa (SSA) recording the highest reduction at 39.7% in 2019 [2]. As a result, the life expectancy of people with HIV (PWH) has substantially improved leading to an increase in the population of PWH over the age of 50 years [3]. This has coincided with an increase in the prevalence of age-related non-communicable diseases (NCDs) such as cardiovascular diseases among PWH [4]. Moreover, it is expected that over time, the prevalence of cardiovascular diseases among PWH in SSA is likely to increase due to the high predisposition [5, 6]. Hypertension is a known and important risk factor of cardiovascular diseases which is highly prevalent among PWH at 23.6–25.2% globally [7, 8] and at 8%—27.2% in Uganda [9–12].

The increasing incidence and prevalence of hypertension among PWH is likely to negatively affect the gains made in improving the health-related quality of life (HRQoL) of PWH through the provision of ART. In fact, the HRQoL of PWH greatly improves after initiation of ART [13] while individuals diagnosed with hypertension have been shown to have worse HRQoL when compared to their normotensive counterparts [14, 15]. Therefore, the possible HIV-hypertension comorbidity could significantly affect HRQoL. This could partly be attributed greater pill burden leading to sub-optimal adherence resulting into virological failure [16, 17]. The synergistic presence of risk factors for cardiovascular diseases such as alcohol use, smoking, and advanced age [18–20] as well as HIV associated stigma and adverse effects of ART [18, 21, 22] could be other probable causes of low HRQoL in this group.

The absence of evidence highlighting the effect of the HIV and hypertension on HRQoL makes the design of better target locally adapted HRQoL improving interventions difficult. This study compared HRQoL in PWH with and without hypertension in Uganda. The Medical Outcome Study (MOS-HIV) score, a disease specific HRQoL tool [23] and the EuroQol-5D-5L (EQ-5D-5L), a generic HRQoL tool [24] which are likely to complement each other to generate HIV specific HRQoL and health utility scores [25] were used.

## Methods

### Study design

We conducted a comparative cross-sectional study between September 2020 and April 2021.

### Study setting

This study was conducted at the Infectious Diseases Institute (IDI) in Kampala, Uganda. The IDI is a center of excellence for HIV care and treatment with over 322,000 PWH under its care and over 70,000 patients recruited into its longitudinal cohorts [26]. IDI maintains an Antiretroviral therapy Long-Term (ALT) cohort which was established in 2014 with participants from urban communities in the central region of Uganda. The ALT is a prospective cohort of 1,000 PWH at the IDI clinic who have been on ART for over 10 years, enrolled at 1 year, and followed up yearly for 10 years. The detail methodology has been published elsewhere [27]. The cohort aims to determine the incidence of long-term drug side effects and toxicity, ART durability, and development of co-morbidities, with an emphasis on NCDs [27].

### Study population

We recruited PWH with or without hypertension who had been on ART for at least 10 years and were in the ALT cohort. The prolonged exposure to certain classes of ART ($\geq$ 10 years) and HIV could increase their risk of developing hypertension hence their selection. Participants with hypertension were entered into this study based on the existing ALT Cohort standard operating procedure. The ALT Cohort standard operating procedures defined hypertension as three systolic measurements above 140 mm Hg or diastolic measurement above 90 mm Hg or history of treatment with antihypertensive drugs (as per ICEA) or patient-reported hypertension [27]. In addition, we included participants who were 18 years and above, were willing to give written informed consent, and willing to complete all study procedures. Pregnant women at the time of the study were excluded.

### Sampling technique

We obtained an electronic register of participants in the ALT cohort at IDI and performed stratified random sampling using computer generated random numbers based on the strata of presence or absence of confirmed hypertension at most recent follow up date for the participants (at most 6 months prior to our study). Hypertension was defined as either taking antihypertensives at the time of the last clinic visit and or elevated blood pressures of >140/90 mmHg as per the Uganda National guidelines at the time. Of these, 174 had a diagnosis of hypertension as determined by the ALT cohort's clinical criteria and 826 did not have a diagnosis of hypertension (Fig 1). Due to smaller numbers of the hypertensive patients, all of them were subsequently considered while due to the large numbers only a subset was considered. Participants' phone contact information were retrieved from the clinic database and research assistants made phone calls with the potential participants to provide study related information and schedule study visits. Stratified random sampling was chosen because it would provide an unbiased representative sample of the different subgroups under consideration for our study.

### Sample size estimation

We calculated the sample size using the formula of comparing means in two groups and a prior study conducted in Rwanda among HIV positive patients where the overall quality of life (irrespective of mental or physical health scores) was estimated at 50.4 ± 31.74 [19]. Due to

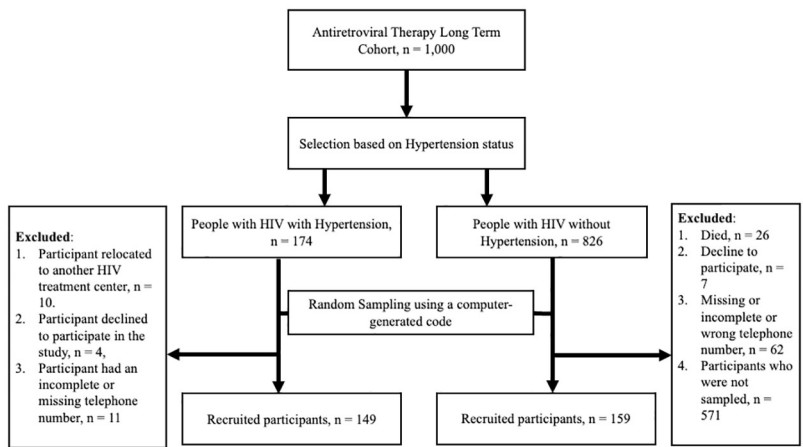

**Fig 1. The flow chart representing how participants were selected in this study.**

absence of data comparing HRQoL among patients with HIV and HTN and those with HIV alone using the EQ-5D-5Lquestionnaire at the time of the study, we assumed a 10-point difference mean quality of life score points on the MOS-HIV questionnaire between the two groups at a power of 80% and type 1 error of 0.05. The more conservative 10-point difference in mean quality of life score points instead of the 16-point (derived from 0.5*SD, SD = 31.74) [28] clinically meaningful difference was chosen to further minimize the type 1 error rate. We needed 318 patients (159 participants in each group), assuming an equal distribution of participants in each group. We recruited 308/318 participants at IDI study site with all the 159 participants without hypertension recruited but 149/159 participants with hypertension were recruited giving us a power of 77.5%. Reasons for failure to reach the desired sample size in the latter category included death, failure to consent, being unreachable as shown in Fig 1.

## Data collection

At enrollment, we conducted face to face interviews with participants who consented to participate in the study using validated English or Luganda versions of the EQ-5D-5L and MOS-HIV questionnaires at a single sitting in a quiet room depending on the participants' preference. Research assistants underwent training using both versions to ensure proper translation and interpretation of the questions. An electronic data collection tool, Kobo Toolbox [29], was used to collect data. Data such as ART duration, WHO clinical stage, viral load, diagnosis of opportunistic infections at the last follow up visit was extracted from the electronic database of the participants. HRQoL was measured using EQ-5D-5L, and disease-specific MOS-HIV tools.

## HRQoL measures

**EQ-5D-5L** [30, 31] is a standardized instrument that measures health status in 5 dimensions–Mobility, Self-Care, Usual Activities, Pain/Discomfort and Anxiety/Depression (EQ-5D). These are combined into a single utility score. The EQ-5D-5L also measures a respondent's self-rated health on a linear visual analogue scale (EQ-VAS 20cm). At the time of the study (mid-year 2020) and analysis [32], there were no value sets available for the Ugandan population and the EQ-5D-5L had not been completely validated in Uganda. Additionally, no specific studies had been conducted in Uganda to evaluate the use of the EQ-5D-5L in chronic diseases

like Hypertension and HIV patients. A single EQ-5D-5L utility score for our participants was computed based on a scoring algorithm from a general population survey in Zimbabwe [33]. The utility scores for the Zimbabwean population were appropriate for use in this valuation of health study because of absence of any other African population utility scores and the closeness of the population characteristics to the Ugandan population. In the foregoing Zimbabwe study, negative values for utility scores worse than death were generated by asking study respondents about how they valued a different set of eight randomly selected health states (either one or two of very mild, mild, moderate, and severe states) and the 33333 as the eighth state. Additionally, in this study, the research evaluated 38 different health states from the combinations of the five EQ-5D domains. The participants ranked states spent in perfect health for ten years as 11111 (example target state) [33] while states worse than death (10- target state) followed by the additional years in full health. Analysis of the self-administered EQ-5D-5L questionnaire and EQ5D visual analogue scale data revealed that nearly one third of participants had some or sever pain/discomfort and few people had problems in self-care, mobility, or usual activities. Higher VAS scores were reported among the Zimbabwean participants. The EQ-5D-5L's performance based on studies done in the *Shona* and English languages in Zimbabwe had been documented with reliability kappa statistics ranging between 0.65 and 1.00 across all domains [33, 34]. In our study, a time trade-off (TTO) approach was used to value health states [30]. Utility scores were got by subtracting one minus disutility based on methods in a study conducted in Ethiopia [35]. The **MOS-HIV** is a 35-item tool that assesses the ten dimensions of heath in individuals living with HIV [36]. It provides overall physical and mental health summary scores that are transformed or standardized to make comparisons among various dimensions that may have different response categories. The MOS-HIV questionnaire has been validated and used in African settings including Rakai in Uganda [19, 37]. The MOS-HIV health survey has 11 dimensions: Health perceptions (5 items), Physical function (6 items); Social function (1 item); Role function (2 items); Cognitive function (4 items); Pain (2 items); Vitality/energy (4 items); Mental health (5 items); Health distress (4 items); Quality of life (1 item); and Health transition (1 item) [23, 38]. The dimensions were represented as health perceptions (Question 1 and 11), bodily pain (question 2 and 3), physical function (question 4), role function (question 5 and 6), social function (question 7), mental health (question 8), vitality/energy (first 4 of question 9), health distress (last 4 of question 9), cognitive function (question 10), quality of life (question 12), and health transition (question 13). The team sought permission to use this tool in the study. Responses were aggregated and converted to a 0–100 point scale, with 100 representing the best health status and 0 the worst state of health as published elsewhere [39].

## Data analysis

Data were cleaned, coded, and analyzed using Stata 15.0. Descriptive statistics such as mean, standard deviations, proportions and interquartile ranges were used to describe the study population. The summary of the participants' responses on a five-point Likert scale of severity for each of the domains of the EQ-5D-5L questionnaire were presented as proportions. Chi square and Fisher's exact tests used to compare values between the non-hypertensive and HIV/hypertension group. The primary outcome was the mean EQ-5D-5L utility score of the study participants. Participant data were manually entered into the EQ-5D-5L five-level crosswalk utility index value calculator as described by the user manual [40, 41]. An EQ-5DL-5L profile was then built for each participant using the calculator algorithm for value sets from the Zimbabwean population [33] and utility scores were calculated based on the Ethiopian population [35]. The HRQoL scores based on the EQ-5D-5L questionnaire were compared by the

different viral load copies of ≤1000 or > 1000 using the Fisher's exact and Mann-Whitney U tests. Questionnaire domains, physical health and mental health summaries were calculated from the MOS-HIV dimensions [42]. The component summary scores (physical and mental health summary scores) were then standardized so that each had a mean of 50 and a standard deviation of 10 [43]. Unadjusted differences in demographic variables and EQ-5D-5L/ MOS-HIV scores between the patients with HIV and hypertension and those without hypertension were assessed using Wilcoxon rank sum tests (for skewed distributions) or student t test (normally distributed) for continuous variables. The Chi square test was used to compare the categorical variables. A linear regression model was fitted by considering the EQ-5D-5L utility scores as the dependent variable. Covariates such as age, sex, hypertension and other social demographic characteristics were then added to the multiple linear regression model as done in a previous study [44]. The models were tested for normality, heteroscedasticity, linearity, and independence of observations. We included factors with a p-value less than 0.20 or clinical significance and known associations from prior literature in the multivariable model [45]. The unadjusted and adjusted coefficients with their 95% confidence intervals were presented.

### Ethics statement

The study was approved by the Makerere University School of Biomedical Sciences Research and Ethics Committee (SBS 750), University of Washington Institutional Review Board (STUDY0000974) and Uganda National Council for Science and Technology approval number HS581ES. The study was done in accordance with the ethical guidelines of the 1975 Declaration of Helsinki and principles of good clinical practice. Participants were informed about the purpose and procedure of the study. Written and verbal informed consent was obtained before enrollment into the study. No participant identifier information was obtained.

## Results

We recruited 308 participants of whom 149 individuals had hypertension and HIV while 159 participants were without hypertension (only had HIV). Fig 1 shows how participants were selected to take part in the study.

### Participant characteristics

Of the 308 participants recruited, 190 (61.7%) were females. Participants with hypertension were older than those without hypertension (Mean ± SD: 53.7± 8.3 years vs 49.9 ± 8.6 years). Table 1 highlights more details on the socio-demographic characteristics of the participants.

Concerning the clinical history of the participants, the median duration of ART was similar for both participants with or without hypertension at 15.35 (14.89–15.79) years and 15.32 (14.94–15.87) years respectively. More participants without hypertension were in stage IV of the WHO HIV staging compared to those with hypertension (41.8% vs 35.1%, respectively). Ninety-nine participants with or without hypertension had a treatment supporter.

Regarding risk factors of cardiovascular disease, only 2 (1.3%) of participants with hypertension had a history of cardiovascular event at their last follow-up. Thirty-five (23.5%) PWH with hypertension had a history of smoking while only 21 (13%) PWH without hypertension had such history. The same number of participants with hypertension or without hypertension had a history of alcohol intake, 24.

**Table 1. The socio-demographic and clinical characteristics of participants recruited into this study.**

| Characteristic | Hypertension, n (%) | No Hypertension, n (%) |
|---|---|---|
| **Age in completed years** | | |
| Mean ± Standard Deviation (SD) | 53.7 ± 8.3 years | 49.9 ± 8.6 years |
| Range | 38–78 | 31–73 |
| **Sex** | | |
| Female | 86 (57.7) | 104 (65.4) |
| Male | 63 (42.3) | 55 (34.6) |
| **Marital status** | | |
| Divorced | 5 (3.4) | 11 (6.9) |
| Married | 51 (34.2) | 46 (28.9) |
| Single | 33 (22.2) | 49 (30.8) |
| Cohabiting | 21 (14.1) | 25 (15.7) |
| Widow/widowed | 39 (26.2) | 28 (17.6) |
| **Occupation** | | |
| None | 18 (12.1) | 18 (11.3) |
| Peasant farmer | 23 (15.4) | 25 (15.7) |
| Housewife | 5 (3.4) | 4 (2.5) |
| Salaried employee | 23 (15.4) | 23 (14.5) |
| Self employed | 74 (49.7) | 80 (50.3) |
| Other (pastor, pensioner) | 6 (4.0) | 9 (5.7) |
| **Monthly income in Uganda shillings in 10^3** | | |
| Median (Interquartile Range (IQR)) | 100 (0–450) | 40 (0–250) |
| **Level of education** | | |
| No formal education | 6 (4.0) | 14 (8.8) |
| Primary | 52 (34.9) | 67 (42.2) |
| Secondary | 69 (46.3) | 58 (36.5) |
| University | 4 (2.7) | 3 (1.9) |
| Other tertiary institution | 18 (12.1) | 17 (10.7) |
| **Duration of ART in years** | | |
| Median (IQR) | 15.35 (14.89–15.79) years | 15.32 (14.94–15.87) years |
| **Latest World Health Organization (WHO) stage** | | |
| I | | |
| II | 19 (12.8) | 17 (10.8) |
| III | 77 (52.0) | 75 (47.5) |
| IV | 52 (35.1) | 66 (41.8) |
| **Presence of treatment supporter** | | |
| No | 50 (33.6) | 54 (34.0) |
| Yes | 99 (66.4) | 99 (66.0) |
| **Viral load at last follow-up visit (copies per ml) [a]** | | |
| ≤1000 | 153 (96.3) | 145 (97.3) |
| > 1000 | 6 (3.8) | 4 (2.7) |
| **Cardiovascular event at last follow-up** | | |
| No | 146 (98.7) | 158 (100.0) |
| Yes | 2 (1.3) | 0 (0.0) |
| **Presence of opportunistic infection at last follow up visit [a]** | | |
| Yes [b] | 0 (0.0) | 2 (1.3) |
| No | 148 (100.0) | 156 (98.7) |
| **Knowledge of the partner's HIV status** | | |

*(Continued)*

**Table 1.** (Continued)

| Characteristic | Hypertension, n (%) | No Hypertension, n (%) |
|---|---|---|
| No | 58 (38.9) | 61 (38.4) |
| Yes | 91 (61.1) | 98 (61.6) |
| **Current use of contraceptives** | | |
| No | 86 (57.7) | 92 (57.9) |
| Yes | 63 (42.3) | 67 (42.1) |
| **History of smoking** | | |
| No | 114 (76.5) | 138 (86.8) |
| Yes | 35 (23.5) | 21 (13.2) |
| **History of alcohol intake** | | |
| No | 125 (83.9) | 135 (84.9) |
| Yes | 24 (16.1) | 24 (15.1) |

[a] Virological failure classification based on the *Consolidated guidelines on the use of antiretroviral drugs for treating and preventing HIV infection: recommendations for a public health approach*. World Health Organization, 2016.

[b] Only opportunistic infection present was tuberculosis.

WHO- World Health Organization, SDA- Seventh Day Adventist.

## HRQoL scores

Based on the EQ-5D-5L utility scores (Table 2), the median utility score from the participants was 0.73 (0.38–0.80). Participants with hypertension had lower median utility scores compared to participants without hypertension (0.71 (0.33–0.80) vs 0.80 (0.44–0.80), p value = 0.029). Close to three quarters of the participants, 223 (72.4%), had moderate problems with mobility with more participants without hypertension reporting this issue than those without hypertension (131 (82.4%) VS 92 (61.7%), p value < 0.001). In addition, a higher proportion of participants with hypertension had extreme problems, 23 (15.4%) as compared to those without hypertension, 8 (5%), p value < 0.001. Fifty-four (17.5%) participants had extreme anxiety/depression. Having hypertension was associated with higher proportions of moderate or severe or extreme problems as compared to those without hypertension, p value < 0.05.

Regarding MOS-HIV scores (Table 3), participants with hypertension had lower mean overall physical health scores compared to participants without hypertension (48.44 ± 10.17 VS 51.44 ± 9.65, p value = 0.008). The overall mean score was lowest in the physical function domain, 18.67 ± 21.39. In general health perception domain, participants without hypertension had a lower mean score than those with hypertension (26.54 ±17.82 Vs 30.76 ± 22.95) but there was no statistical significance (p = 0.072). Role functioning dimension had the biggest point difference in the physical health domain between the two sets of participants (9.66, 95% Confidence Interval (CI) (2.62–16.71)), with participants with hypertension having a lower score in this domain than those without hypertension (75.36 ± 33.75 VS 85 ± 29.03, p value = 0.007). In the mental health domain, the overall mean mental health summary score was 50 ± 10 with participants with hypertension having a higher overall mean mental health summary score (50.62 ± 9.75) than those without hypertension (49.42 ± 10.21). Generally, participants with hypertension had better scores than those without hypertension in majority of the components, however, none of the components were statistically significant.

**Factors associated with low HRQoL scores based on EQ-5D-5L utility scores.** Based on a multiple linear regression (Table 4), having an income > 70,000 Uganda shillings was associated with a 0.05 units reduction in HRQoL scores as compared to those earning less ≤ 70,000

**Table 2. The proportion of participants who had different HRQoL scores based on the EQ-5D-5L questionnaire.**

| Characteristic | Overall*, n (%) | Hypertension, n (%) | No Hypertension, n (%) | P value |
|---|---|---|---|---|
| **Mobility** | | | | |
| No problem | 2 (0.6) | 0 (0.0) | 2(1.3) | <0.001** |
| Slight problem | 41 (13.3) | 26 (17.4) | 15 (9.4) | |
| Moderate problem | 223 (72.4) | 92 (61.7) | 131(82.4) | |
| Severe problem | 11 (3.6) | 3 (1.9) | 8 (5.4) | |
| Extreme problem | 31 (10.1) | 23 (15.4) | 8 (5.0) | |
| **Self care** | | | | |
| No problem | 1 (0.3) | 0 (0.0) | 1 (0.6) | 0.153** |
| Slight problem | 8 (2.6) | 2 (1.3) | 6 (3.8) | |
| Moderate problem | 290 (94.2) | 140 (94.0) | 150 (94.3) | |
| Severe problem | 2 (0.7) | 2 (1.3) | 0 (0.0) | |
| Extreme problem | 7 (2.3) | 5 (3.4) | 2 (1.3) | |
| **Usual activities** | | | | |
| No problem | 5 (1.6) | 2 (1.3) | 3 (1.9) | 0.574** |
| Slight problem | 20 (6.5) | 11(7.4) | 9 (5.7) | |
| Moderate problem | 270 (87.7) | 127 (85.2) | 143 (89.9) | |
| Severe problem | 3 (1.0) | 2 (1.3) | 1 (0.6) | |
| Extreme problem | 10 (3.2) | 7 (4.7) | 3 (1.9) | |
| **Pain/discomfort** | | | | |
| No problem | 6 (1.9) | 2 (1.3) | 4 (2.5) | 0.156** |
| Slight problem | 63 (20.4) | 38 (25.5) | 25 (15.7) | |
| Moderate problem | 157 (51.0) | 67 (45.0) | 90 (56.6) | |
| Severe problem | 23 (7.5) | 11(7.4) | 12 (7.6) | |
| Extreme problem | 59 (19.2) | 31 (20.8) | 28 (17.6) | |
| **Anxiety/Depression** | | | | |
| No problem | 186 (60.4) | 76 (51.0) | 110 (69.2) | 0.007** |
| Slight problem | 6 (1.9) | 4 (2.7) | 2 (1.3) | |
| Moderate problem | 50 (16.2) | 34 (22.8) | 16 (10.1) | |
| Severe problem | 12 (3.9) | 7 (4.7) | 5 (3.1) | |
| Extreme problem | 54 (17.5) | 28 (18.8) | 26 (16.4) | |
| **EQ5D5L utility index scores***, median and Interquartile range** | 0.73 (0.38–0.80) | 0.71 (0.33–0.80) | 0.80 (0.44–0.80) | 0.029***** |
| **Visual analog scale (VAS) score****, median and Interquartile range** | 80 (65–90) | 75 (60–90) | 80 (70–90) | 0.241***** |

*Overall (entire dataset of 308 is considered without categorizations)

**P value based on Fisher's exact test

*** EQ-5D-5L utility index ranges from 0 (worst HRQOL) to 1 (best HRQOL)

****VAS score ranges from 0 = worst health to 100 = best Health

*****P value based on the Mann Whitney U test because of unequal variances.

Uganda shillings (Adjusted β = -0.049, 95% CI, -0.097—(-0.001), p value = 0.044). The presence of a history of hypertension led to 0.040 reduction in the HRQoL utility scores as compared to those without hypertension (Adjusted β = -0.040, 95% CI -0.074—(-0.005), p value = 0.023). Participants who had disclosed their HIV status to their partners had a 0.065 reduction in HRQoL scores as compared to those who had not (Adjusted β = -0.065, 95% CI, -0.122—(-0.008). Those participants who had a history of smoking had a 0.053 reduction in the HRQoL scores as compared to those who had no history of smoking (Adjusted β = -0.053,

**Table 3. The mean scores of HRQoL scores based on MOS-HIV.**

| MOSHIV dimensions | Overall, mean ± SD | Minimum | Maximum | Hypertension, mean ± SD | No hypertension, mean ± SD | Mean difference between Hypertension and non-Hypertension (95% CI) | Value*** |
|---|---|---|---|---|---|---|---|
| **Physical health summary** | **50 ± 10** | **15.02** | **62.06** | **48.44 ± 10.17** | **51.44 ± 9.65** | **3.01 (0.78–5.23)** | **0.008** |
| General health perceptions | 28.57 ± 20.52 | 0 | 84.21 | 30.76 ± 22.95 | 26.54 ± 17.82 | -4.21 (-8.80–0.37) | 0.072 |
| Physical functioning | 18.67 ± 21.39 | 0 | 91.67 | 20.66 ± 20.95 | 16.82 ± 21.71 | -3.84 (-8.63–0.95) | 0.116 |
| Role functioning | 80.36 ± 31.71 | 0 | 100 | 75.36 ± 33.75 | 85.00 ± 29.03 | 9.66 (2.62–16.71) | **0.007** |
| Social functioning | 73.25 ± 22.39 | 0 | 100 | 73.48 ± 24.13 | 73.63 ± 20.73 | 0.79(-4.24–5.82) | 0.756 |
| Bodily pain | 52.38± 18.80 | 0 | 100 | 50.52± 19.19 | 54.09 ± 18.32 | 3.57 (-0.63–7.78) | 0.096 |
| **Mental health summary** | **50 ± 10** | **13.61** | **70.64** | **50.62 ± 9.75** | **49.42 ± 10.21** | **-1.19 (-3.44–1.05)** | **0.295** |
| Cognitive functioning | 69.62 ± 18.95 | 0 | 100 | 71.58 ± 16.75 | 67.81 ± 20.68 | -3.78 (-8.01–0.46) | 0.081 |
| Mental health | 63.35± 17.10 | 0 | 100 | 65.02 ± 18.17 | 61.80 ± 15.94 | -3.22 (-7.05–0.59) | 0.098 |
| Vitality/energy/fatigue | 59.14 ± 18.45 | 0 | 100 | 59.89 ± 18.58 | 58.43 ± 18.54 | -1.46 (-5.63–2.70) | 0.491 |
| Health distress | 71.93 ± 17.59 | 0 | 100 | 70.37 ± 20.00 | 73.38 ± 14.94 | 3.00 (-0.94–6.94) | 0.135 |
| Quality of life | 36.61 ± 31.66 | 0 | 100 | 34.79 ± 30.39 | 38.28 ± 32.81 | 3.48 (-3.62–10.59) | 0.335 |
| Health transitions | 28.41 ± 31.01 | 0 | 100 | 26.52 ± 31.03 | 30.16 ± 30.98 | 3.63 (-3.32–10.59) | 0.305 |

***P values based on comparing the means between the two different groups.

95% CI, -0.100—(-0.005), p value = 0.029). The 0.05 reduction may have uncertain significance to the participants.

## Discussion

In this cross-sectional study, we found that hypertensive participants had lower mean utility scores than those without hypertension based on EQ-5D-5L tool. Whereas participants with hypertension had better mental health scores than their counterparts, their physical health scores were worse than those without hypertension. After adjusting for confounders, hypertension, higher income status, disclosure of the HIV status of the participants to their participants and history of smoking were associated with low EQ-5D-5L utility scores.

There is limited literature comparing the health-related quality of life of HIV positive individuals with hypertension and those without. However, studies assessing the quality of life of PWH have shown that PWH on ART have improved quality of life [19, 46] which is consistent with our findings. On the other hand, participants with hypertension had lower scores in most of the components especially in the physical health summary components which is similar to what was reported in other settings [19, 47]. Although there was no significant difference in the mental health scores in both groups, participants had lower mental health scores overall. This has also been reported in Rwanda [19], Uganda [37] and Ethiopia [47].

Our findings suggest that hypertension reduces the quality of life among PWH. Schenker and colleagues showed that the presence of chronic co-morbidities as well as polypharmacy in individuals with advanced age led to poor quality of life and increased symptom burden [48]. In our study, individuals with hypertension were slightly older than those without and had poorer physical health scores which could have had a negative effect on their ability to carry out their day-to-day activities leading to the differences in the physical health summary scores,

**Table 4. The factors associated with low HRQoL scores based on EQ-5-D5L utility scores.**

| Characteristic | Un adjusted coefficients, β (95% Confidence intervals) | P value | Adjusted coefficients, β (95% Confidence intervals) | P value |
|---|---|---|---|---|
| **Hypertension** | | | | |
| No | Reference | | Reference | |
| Yes | -0.042(-0.076—(-0.007)) | 0.017 | -0.040 (-0.074—(-0.005)) | 0.023 |
| **Marital** | | | | |
| Married | Reference | | | |
| Single | 0.036(-0.006–0.077) | 0.095 | | |
| Widowed | 0.023 (-0.021–0.068) | 0.305 | | |
| Divorced | -0.034(-0.114–0.046) | | | |
| **Education level** | | | | |
| Primary or no formal education | Reference | | | |
| Secondary | -0.019(-0.056–0.018) | 0.321 | | |
| University or tertiary | -0.017 (-0.071–0.036) | 0.514 | | |
| **Treatment supporter** | | | | |
| No | Reference | | Reference | |
| Yes | 0.012(-0.024–0.049) | 0.497 | 0.015(-0.021–0.053) | 0.405 |
| **Income status per month (in Uganda Shillings X10^3)** | | | | |
| ≤70 | Reference | | Reference | |
| 71–210 | -0.017(-0.001–0.076) | 0.499 | -0.049(-0.097—(-0.001)) | 0.044 |
| >210 | 0.037(-0.069–0.034) | 0.058 | -0.043(-0.084—(-0.002)) | 0.038 |
| **Sex** | | | | |
| Male | Reference | | Reference | |
| Female | -0.007(-0.042–0.029) | 0.711 | -0.043(-0.085–0.000) | 0.050 |
| **Duration of ART** | | | | |
| >15 year | Reference | | | |
| ≤15 year | -0.001(-0.039–0.037) | 0.953 | | |
| **Do you know your partners HIV status** | | | | |
| No | Reference | | Reference | |
| Yes | 0.004 (-0.032–0.039) | 0.839 | -0.51(-0.007–0.108) | 0.086 |
| **Has the patient disclosed their HIV status to partner** | | | | |
| No | Reference | | Reference | |
| Yes | -0.018(-0.053–0.017) | 0.314 | -0.065(-0.122—(-0.008)) | 0.026 |
| **Latest WHO stage** | | | | |
| II | Reference | | | |
| III | 0.014(-0.042–0.071) | 0.617 | | |
| IV | 0.01(-0.05–0.06) | 0.849 | | |
| **Age** | -0.001 (-0.005–0.003) | 0.509 | | |
| **Currently smoke** | | | | |
| No | Reference | | Reference | |
| Yes | -0.051(-0.096—(-0.007)) | 0.024 | -0.053(-0.100—(-0.005)) | 0.02 |
| **Currently drink alcohol** | | | | |
| No | Reference | | | |
| Yes | -0.029 (-0.076–0.018) | 0.235 | | |

mobility, and anxiety/depression scores. Another study done in Malawi found that hypertensive PWH had poor adherence to their anti-hypertensive medication, difficulty in adjusting their lifestyles, and the costs involved in purchasing the medicine, which may have further contributed to poor HRQoL scores. Whereas mental health scores were not statistically significant, they were generally low–possibly due to the high prevalence of psychological disorders in HIV positive individuals and the social stigma they face [49, 50]. In addition, studies have shown that having a treatment supporter is associated with improved quality of life especially mental health through the provision of social support [51]. Much as the study was done among participants who have been on long term ART (> 10 years) and almost all participants had treatment supporters, our study found that the disclosure of HIV/AIDS status to partners was associated with low HRQoL scores. This could be attributed to the rejection they faced or are still facing from their partners [52] causing further distress. Therefore, subsequent studies should explore how stigma and disclosure of HIV status could affect HRQoL of HIV positive individuals who have been on long term ART.

Co-morbidity with hypertension, history of smoking, and higher income status was associated with low HRQoL. Participants with hypertension have a high risk of developing cardiovascular diseases which is likely to cause significant morbidity including limitations in performing their daily physical intensive activities hence the poor score in the role functioning domain. Unlike a study done in China [53], a higher income status was associated with low HRQoL scores in this study. This could be due to stress from the closure of businesses and distortion in economic activities because of the coronavirus disease lockdown. The lockdown restriction of movements prevented people with chronic diseases such as HIV and hypertension from accessing their clinics, leading to disruptions in the supply of critical medications such as ART and antihypertensives [54]. This situation, coupled with financial disruptions, adversely affected mental health, increasing the risk of anxiety and depression in PWH with or without hypertension. History of smoking was associated with a reduction in HRQoL which may be explained by the increase in evidence suggesting that smoking has a negative impact on the psychological wellbeing of individuals and less likely to seek social support and have poor adherence to ART as well as more likely to experience more social stigma and develop other co-morbidities such as liver cirrhosis and cancers hence affecting their HRQoL [55].

Our study has limitations. First, the EQ-5D-5L has not been validated in a Ugandan population. Secondly, we were unable to collect data on co-morbidities such as obesity, diabetes, hypertension treatment and pocket costs that could have impacted results. Therefore, future studies should explore how these factors affect the HRQoL of PWH with or with hypertension. Thirdly, there was a significant age difference between PWH with hypertension and those without. Lastly, the study was conducted among patients who have been on ART ≥10 years receiving care from a center of excellence in HIV care where they receive high quality care. This population may not be representative of all individuals living with HIV. Despite these limitations, the proportion of ageing PWH in Uganda is progressively increasing which highlights the need to evaluate the changes in functionality, monitor treatment outcomes and effects on patients, and changes in care because of multimorbidity. Future longitudinal studies should explore the differences and changes in HRQoL scores of PWH with co-morbidities in rural and urban settings due to the differences in lifestyles.

## Conclusion

Our findings provide evidence on the potential effect of hypertension on the HRQoL of PWH and illustrate the need for strategies targeting the improvement of the HRQoL of PWH on long term ART with or without hypertension. Further research on how HRQoL affects

treatment outcomes of PWH diagnosed with hypertension compared to those without hypertension is warranted.

HIV/AIDS programs should incorporate hypertension-related screening services into routine clinic as well as strengthen patient education on hypertension and its risk factors to foster early management of hypertension.

## Acknowledgments

We thank ALT cohort study team for the assistance render to us during data collection. We are grateful to the participants in the ALT cohort who shared their time to come and meet us during the data collection despite COVID-19 pandemic associated restrictions.

## Author Contributions

**Conceptualization:** Charles Batte, John Mukisa, Jasper Nidoi, Barbara Castelnuovo, Mohammed Lamorde, David Meya, William Checkley, Robert Kalyesubula, Trishul Siddharthan, Joseph B. Babigumira.

**Data curation:** Charles Batte, Andrew Weil Semulimi, Mariam Nakabuye, Rosalind Parkes Ratanshi, Barbara Castelnuovo, Trishul Siddharthan.

**Formal analysis:** Andrew Weil Semulimi, John Mukisa, Mariam Nakabuye, David Mukunya, David Meya, William Checkley, Joseph B. Babigumira.

**Funding acquisition:** Charles Batte.

**Investigation:** Charles Batte, John Mukisa, Mariam Nakabuye, Jasper Nidoi, David Mukunya, William Checkley, Joseph B. Babigumira.

**Methodology:** Charles Batte, Andrew Weil Semulimi, John Mukisa, Mariam Nakabuye, Jasper Nidoi, David Mukunya, Rosalind Parkes Ratanshi, Barbara Castelnuovo, Mohammed Lamorde, Robert Kalyesubula, Trishul Siddharthan, Joseph B. Babigumira.

**Project administration:** Charles Batte, Andrew Weil Semulimi, Mariam Nakabuye.

**Resources:** John Mukisa.

**Software:** Charles Batte, Andrew Weil Semulimi, John Mukisa.

**Supervision:** Charles Batte, John Mukisa, David Mukunya, Rosalind Parkes Ratanshi, Barbara Castelnuovo, Mohammed Lamorde, David Meya, William Checkley, Trishul Siddharthan, Joseph B. Babigumira.

**Validation:** Charles Batte, Andrew Weil Semulimi, John Mukisa, Mariam Nakabuye, Jasper Nidoi, David Mukunya, Robert Kalyesubula.

**Visualization:** Charles Batte, John Mukisa, Mariam Nakabuye, Jasper Nidoi, David Mukunya, Rosalind Parkes Ratanshi, Barbara Castelnuovo, Mohammed Lamorde, David Meya, William Checkley, Robert Kalyesubula, Trishul Siddharthan, Joseph B. Babigumira.

**Writing – original draft:** Charles Batte, Andrew Weil Semulimi, John Mukisa.

**Writing – review & editing:** Charles Batte, Andrew Weil Semulimi, John Mukisa, Mariam Nakabuye, Jasper Nidoi, David Mukunya, Rosalind Parkes Ratanshi, Barbara Castelnuovo, Mohammed Lamorde, David Meya, William Checkley, Robert Kalyesubula, Trishul Siddharthan, Joseph B. Babigumira.

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
