## [Decision Letter · Decision Letter 0]

6 May 2024

PONE-D-24-03016Health-related quality of life and hypertension in people with HIV on long-term antiretroviral therapy in Uganda.PLOS ONE

Dear Dr. Semulimi,

Thank you for submitting your manuscript to PLOS ONE. After careful consideration, we feel that it has merit but does not fully meet PLOS ONE’s publication criteria as it currently stands. Therefore, we invite you to submit a revised version of the manuscript that addresses the points raised during the review process.

We look forward to receiving your revised manuscript.

Kind regards,

Omar Sued, MD, PhD

Academic Editor

PLOS ONE

Journal Requirements:

This project was supported by NIH Research Training Grant # D43 TW009345 funded (CB) by the Fogarty International Center, the NIH Office of the Director Office of AIDS Research, the NIH Office of the Director Office of Research on Women's Health, The National Heart, Lung and Blood Institute, the National Institute of Mental Health, and the National Institute of General Medical Sciences. The content is solely the responsibility of the authors and does not necessarily represent the official views of the National Institute of Health.

The ALT cohort (BC) is funded by Janssen, the pharmaceutical company of Johnson & Johnson, through a grant to the Academy for Health Innovation, Uganda at Infectious Diseases Institute. The funders had no role in study design, data collection and analysis, decision to publish, or preparation of the manuscript.

Additional Editor Comments:

Dear Editor

Thank you for the opportunity to review this revised version of this study. Hypertension is becoming a more frequent condition among HIV individuals due to the aging of population and the additive value of NCDs. Quality of life related to these conditions are not usually reported, which increase the value of this study.

Below are my comments, with the comments of other reviewers compiled.

1) In the abstract the utility score phrase is repeated

2) Please ensure the access to the data are working (figshare: http://10.0.23.196/m9.figshare.20014799)

3) Clarify the inclusion criteria for HTN patients> in Line 105 it is said to be at least 3 records >140/90 but in line 115 it is said to be diagnosed people, receiving HTN treatment or >140/90. In addition, the rational for selecting only people +10 years on ART is unclear and not explained in the text.

4) Line 118 seems incomplete

5) Line 127 seems a typo in " EQ5D-5Lquestionnaire"

6) Line 132: It is suggested to change 'assuming' to 'assuming an equal distribution of participants in each group' for greater clarity."

7) Line 150: In the following text, there is an extra dot after '2020': 'At the time of the study (mid-year 2020.) and analysis...'"

8) Line 163: I think there is a discrepancy in the name of the questionnaire in the text. It reads E5Q-5D-5L, but I believe it should be EQ-5D-5L.

9) Line 189: The word "hypertension" should be in lowercase.

10) Line 198:"In the word 'differences', there is an extra space."diffe-rences"

11) Line 219-220: The wording can be improved as follows "More participants without hypertension were in stage IV of the WHO HIV staging compared to those with hypertension (41.8% vs 35.1%, respectively)."

Discussion

12) Line 290 propose those who disclosed their status had lower QoL because discrimination, but more data is needed for this, as those who do not disclose could have higher internal stigma, other potential causes should be identified for this situation

13) Does patients with HTN have integrated care? or they need to go more frequently to the hospital or laboratory services? How many pills they had to take, are the HTN pills provided for free? Those factors could impair QoL and were not mentioned.

14) COVID19 was estimated to affect QoL in people with higher income, as was at this time the study happended. This suggest that authors should provide a clearer contextual information about how COVID19 could have impacted all the other dimensions of QoL among people with or without HTN, it is that HTN services were closed? etc

Limitations

15) The authors should state important limitations

a) No validated EuroQL for Uganda

b) Significant age difference with the comparator

c) Population no representative of all HIV (+10 year retained in services)

b) No data on obesity, diabetes, HTN treatment and pocket costs that could have impacted results.

Reviewers' comments:

Reviewer's Responses to Questions

**Comments to the Author**

1. Is the manuscript technically sound, and do the data support the conclusions?

Reviewer #1: Yes

Reviewer #2: Yes

2. Has the statistical analysis been performed appropriately and rigorously? 

Reviewer #1: Yes

Reviewer #2: I Don't Know

3. Have the authors made all data underlying the findings in their manuscript fully available?

Reviewer #1: Yes

Reviewer #2: Yes

4. Is the manuscript presented in an intelligible fashion and written in standard English?

Reviewer #1: Yes

Reviewer #2: Yes

5. Review Comments to the Author

**Reviewer #1:** I don't have any comments. I think the call is very pertinent the calls for inclusion of quality-of-life assessment in the management of PWH who have been diagnosed with hypertension to identify those at risk and plan early interventions.

**Reviewer #2: **In this study, Charles Batte and his colleagues explore the impact on the quality of life of people living with HIV and hypertension. Overall, I consider it a valuable work that provides important information. However, I believe it would be beneficial to include additional information about the antiretroviral regimens received by the participants. This is a relevant topic, especially because it has been suggested that integrase inhibitors (INSTIs) may be associated with a potential increase in blood pressure. Additionally, it would be pertinent to discuss the possible impact on the quality of life of these regimens due to factors such as weight gain and the risk of developing new conditions such as diabetes and/or hypertension. I suggest reviewing the provided link to access additional information, as I was unable to view the data when attempting to access it directly through the browser or by accessing figshare: http://10.0.23.196/m9.figshare.20014799.

The rest of my observations are minor.

Line 127:There is a missing space in the questionnaire name " EQ5D-5Lquestionnaire"

Line 132: It is suggested to change 'assuming' to 'assuming an equal distribution of participants in each group' for greater clarity."

Line 150: In the following text, there is an extra dot after '2020': 'At the time of the study (mid-year 2020.) and analysis...'"

Line 163: I think there is a discrepancy in the name of the questionnaire in the text. It reads E5Q-5D-5L, but I believe it should be EQ-5D-5L.

Line 189: The word "hypertension" should be in lowercase.

Line 198:"In the word 'differences', there is an extra space."diffe-rences"

Line 219-220: The wording can be improved as follows "More participants without hypertension were in stage IV of the WHO HIV staging compared to those with hypertension (41.8% vs 35.1%, respectively)."

6. PLOS authors have the option to publish the peer review history of their article (what does this mean?). If published, this will include your full peer review and any attached files.

Reviewer #1: **Yes: **Nancy Virginia Sandoval Paiz

Reviewer #2: **Yes: **Rodríguez-Aldama Juan Carlos

---

## [Author Response · Author response to Decision Letter 0]

5 Jun 2024

Thank you for the comments. These have been helpful in improving the manuscript.

Response to the Comments

1. In the abstract the utility score phrase is repeated.

This has been revised. We have removed the repeated phrase. 

2. Please ensure the access to the data are working (Figshare: http://10.0.23.196/m9.figshare.20014799)

This has been updated and the access to the raw data is now open to the public.

3. Clarify the inclusion criteria for HTN patients> in Line 105 it is said to be at least 3 records >140/90 but in line 115 it is said to be diagnosed people, receiving HTN treatment or >140/90. In addition, the rational for selecting only people +10 years on ART is unclear and not explained in the text.

This has been revised and updated line 106 to 112. It reads as follow: We recruited PWH with or without hypertension who had been on ART for at least 10 years and were in the ALT cohort. The prolonged exposure to certain classes of ART (≥ 10 years) and HIV could increase their risk of developing hypertension hence their selection. Participants with hypertension were entered into this study based on the existing ALT Cohort standard operating procedure. The ALT Cohort standard operating procedures defined hypertension as three systolic measurements above 140 mm Hg or diastolic measurement above 90 mm Hg or history of treatment with antihypertensive drugs (as per ICEA) or patient-reported hypertension

4. Line 118 seems incomplete

This has been revised.

5. Line 127 seems a typo in " EQ5D-5Lquestionnaire"

This has been revised.

6. Line 132: It is suggested to change 'assuming' to 'assuming an equal distribution of participants in each group' for greater clarity."

This has been revised. 

7. Line 150: In the following text, there is an extra dot after '2020': 'At the time of the study (mid-year 2020.) and analysis...'"

This has been revised.

8. Line 163: I think there is a discrepancy in the name of the questionnaire in the text. It reads E5Q-5D-5L, but I believe it should be EQ-5D-5L.

This has been corrected.

9. Line 189: The word "hypertension" should be in lowercase.

This has been corrected.

10. Line 198:"In the word 'differences', there is an extra space."diffe-rences"

This has been corrected.

11. Line 219-220: The wording can be improved as follows "More participants without hypertension were in stage IV of the WHO HIV staging compared to those with hypertension (41.8% vs 35.1%, respectively)."

This has been revised and corrected, line 217-218.

Discussion

12. Line 290 propose those who disclosed their status had lower QoL because discrimination, but more data is needed for this, as those who do not disclose could have higher internal stigma, other potential causes should be identified for this situation.

It is true that those who did not disclose their HIV status could experience internalised stigma which could ultimately affect their HRQoL, and this is a valid concern. We believe that this is a complex question that cannot be addressed in the present study and would require a robust research study to understand the dynamics involved between disclosure, stigma and health outcomes in PWH. This has been stated a future research area for the scientific community to explore, line 385 and 386.

13. Do patients with HTN have integrated care? or they need to go more frequently to the hospital or laboratory services? How many pills they had to take, are the HTN pills provided for free? Those factors could impair QoL and were not mentioned.

We recognise that pill burden, cost of pills as well as inconveniences in accessing care could impart HRQoL. However, this was not explored in this study and has been included as a limitation.

14. COVID19 was estimated to affect QoL in people with higher income, as was at this time the study happened? This suggest that authors should provide a clearer contextual information about how COVID19 could have impacted all the other dimensions of QoL among people with or without HTN, it is that HTN services were closed? Etc

This has been revised and updated illustrating how the restriction imposed during the COVID-19 could have affect the HRQoL of PWH with or with hypertension, line 320-323.

Limitations

15. The authors should state important limitations

a) No validated EuroQol for Uganda

b) Significant age difference with the comparator

c) Population no representative of all HIV (+10 year retained in services)

b) No data on obesity, diabetes, HTN treatment and pocket costs that could have impacted results.

The limitations have been revised to reflect this, line 329- 342.

Reviewer #1: I don't have any comments. I think the call is very pertinent the calls for inclusion of quality-of-life assessment in the management of PWH who have been diagnosed with hypertension to identify those at risk and plan early interventions.

Thank you for the comment.

Reviewer #2: In this study, Charles Batte and his colleagues explore the impact on the quality of life of people living with HIV and hypertension. Overall, I consider it a valuable work that provides important information. However, I believe it would be beneficial to include additional information about the antiretroviral regimens received by the participants. This is a relevant topic, especially because it has been suggested that integrase inhibitors (INSTIs) may be associated with a potential increase in blood pressure. Additionally, it would be pertinent to discuss the possible impact on the quality of life of these regimens due to factors such as weight gain and the risk of developing new conditions such as diabetes and/or hypertension. I suggest reviewing the provided link to access additional information, as I was unable to view the data when attempting to access it directly through the browser or by accessing figshare: http://10.0.23.196/m9.figshare.20014799.

The rest of my observations are minor.

---

## [Editor Report · Decision Letter 1]

26 Jun 2024

Health-related quality of life and hypertension in people with HIV on long-term antiretroviral therapy in Uganda.

PONE-D-24-03016R1

Dear Dr. Semulimi,

We’re pleased to inform you that your manuscript has been judged scientifically suitable for publication and will be formally accepted for publication once it meets all outstanding technical requirements.

Kind regards,

Omar Sued, MD, PhD

Academic Editor

PLOS ONE

Additional Editor Comments (optional):

Please check the data sharing link is working. It was revised but I could not open it.

All the other concerns were addressed by the authors.
---

## [Editor Report · Acceptance letter]

15 Jul 2024

PONE-D-24-03016R1 

PLOS ONE

Dear Dr. Semulimi, 

I'm pleased to inform you that your manuscript has been deemed suitable for publication in PLOS ONE. Congratulations! Your manuscript is now being handed over to our production team.

Kind regards, 

on behalf of

Dr. Omar Sued 

Academic Editor

PLOS ONE